# Genetic Panel Testing for Malignant Hyperthermia in Japan: Discovery of Novel Variants and Clinical Implications

**DOI:** 10.3390/genes16080944

**Published:** 2025-08-11

**Authors:** Hirotsugu Miyoshi, Keiko Mukaida, Sachiko Otsuki, Kenshiro Kido, Ayako Sumii, Tsuyoshi Ikeda, Guoqiang Xia, Yuko Noda, Tomomi Ishii, Satoshi Kamiya, Soshi Narasaki, Hiroshi Niinai, Yasuo M. Tsutsumi

**Affiliations:** Department of Anesthesiology and Critical Care, Hiroshima University, Hiroshima 34-8551, Japan; mukaida.keiko1@gmail.com (K.M.); sachi85@hiroshima-u.ac.jp (S.O.); kidoken46@gmail.com (K.K.); batakobatake@gmail.com (A.S.); shiyotsudakei@gmail.com (T.I.); d243940@hiroshima-u.ac.jp (G.X.); nodananoda0724@yahoo.co.jp (Y.N.); tomotomo631201@gmail.com (T.I.); satobo-kamiya@hiroshima-u.ac.jp (S.K.); niinai@hiroshima-u.ac.jp (H.N.); yasuo223@hiroshima-u.ac.jp (Y.M.T.)

**Keywords:** malignant hyperthermia, gene panel test, RYR1, CACNA1S, TRPV1

## Abstract

Background: Malignant hyperthermia (MH) is a pharmacogenetic disorder of skeletal muscle triggered by certain anesthetic agents. While Ryanodine Receptor 1 (*RYR1*) and Calcium Voltage-Gated Channel Subunit Alpha1 S (*CACNA1S*) are well-established susceptibility genes, the complete genetic basis of MH remains unclear, particularly in Asian populations. Methods: We conducted gene panel testing targeting 24 calcium-related genes in 338 individuals from 247 Japanese families with suspected or confirmed MH. Variants were analyzed on a gene-by-gene basis, and their pathogenicity was assessed using in silico prediction tools. Additionally, patients were classified into subgroups based on the results of the calcium-induced calcium release (CICR) assay and the Clinical Grading Scale (CGS) score. Results: Candidate pathogenic variants were identified in 118 families (48.2%), including 73 (29.8%) in *RYR1*, 16 (6.5%) in *CACNA1S*, and 62 (25.3%) in other genes. Among CICR-positive families, *RYR1* and *CACNA1S* variants were detected in 42.0% and 5.3% of cases, respectively. In individuals with high CGS scores (Ranks 5–6), *RYR1* and *CACNA1S* variants were observed in 56.0% and 12.0%, respectively. Variants in other genes such as *STAC3*, *CASQ1*, *ATP2A1*, *ASPH*, *HRC* and *TRPV1* were also detected. Conclusions: Our findings confirm the predominant role of RYR1 and CACNA1S in MH susceptibility in the Japanese population and highlight additional candidate genes that may contribute to the condition. Broader genetic screening and functional validation studies are warranted to further elucidate the polygenic nature of MH.

## 1. Introduction

Malignant hyperthermia (MH) is an inherited skeletal muscle disorder that manifests as a severe anesthetic complication, primarily triggered by exposure to volatile inhalational anesthetics or depolarizing muscle relaxants [1,2]. It follows an autosomal dominant inheritance pattern, where individuals with a genetic predisposition develop clinical symptoms upon exposure to specific anesthetic agents. The incidence of MH varies by region: in North America, it is approximately 1.1 per 100,000 anesthetic procedures; in Europe, rates have been reported as low as 1 per 250,000; and in Asia, estimates are approximately 1.37 per 100,000 anesthetics [3,4,5]. The Ryanodine Receptor 1 (*RYR1*) gene (located at 19q13.2) encodes the ryanodine receptor type 1, a calcium release channel located on the sarcoplasmic reticulum of skeletal muscle cells, which plays a central role in excitation–contraction coupling by releasing calcium into the cytoplasm in response to depolarization [6]. The Calcium Voltage-Gated Channel Subunit Alpha1 S (*CACNA1S*) gene (located at 1q32.1) encodes the alpha-1S subunit of the dihydropyridine receptor (DHPR), a voltage-sensing calcium channel on the transverse tubule membrane that physically interacts with RYR1 to initiate calcium release [7]. Pathogenic variants in RYR1 or CACNA1S can cause abnormal calcium release or sustained calcium leak from the sarcoplasmic reticulum, leading to excessive intracellular calcium levels. Dysregulation of intracellular calcium homeostasis in skeletal muscle cells, mainly due to mutations in RYR1 or CACNA1S, leads to a cascade of pathological events, including muscle rigidity, hypermetabolism, hyperthermia, and life-threatening arrhythmias [3,8,9]. The administration of dantrolene has significantly reduced the mortality rate of MH. However, MH remains a critical and potentially fatal condition, making effective prevention and early diagnosis essential [10,11].

Avoiding exposure to triggering agents is key to preventing MH episodes in genetically susceptible individuals. However, genetic screening is not universally implemented, and research on MH-related genetic variants remains limited, particularly in Asian populations. In contrast, in North America and Europe, genetic testing is widely used as a screening tool, with Diagnostic MH Mutations selected based on established criteria [12,13,14]. This approach has proven beneficial in identifying individuals at risk and guiding clinical management [14]. MH is a very rare disease, and population studies with genetic analysis are valuable.

Among the known MH-related genes, mutations in *RYR1* are the most significant, being identified in 60–86% of MH-susceptible individuals [15,16,17,18,19]. However, a substantial proportion of patients lack identifiable *RYR1* mutations, suggesting the involvement of additional genetic factors [20]. Recent studies have highlighted the role of other genes, including *CACNA1S*, *STAC3*, and potential novel candidates, in MH susceptibility [21,22]. Expanding genetic analyses beyond *RYR1* is therefore essential to fully elucidate the genetic basis of MH susceptibility. In addition, some myopathies [23,24], postoperative rhabdomyolysis [25], exertional rhabdomyolysis/recurrent rhabdomyolysis [26], idiopathic hyperCKemia [27], and exertional heat stroke [28] have been reported to be associated with MH and are therefore subjects of testing for predisposing factors for MH.

The aim of this study is to identify novel genetic variants associated with MH susceptibility. Using targeted gene panel sequencing, we analyzed the distribution of *RYR1*, *CACNA1S*, and unknown genes in individuals suspected of having MH.

## 2. Materials and Methods

### 2.1. Study Design and Patient Selection

This genetic epidemiology study was conducted as a retrospective study analyzing existing data at Hiroshima University. Patients and/or their families with a suspected history of malignant hyperthermia or suspected malignant hyperthermia were included in our study, which enrolled individuals who underwent testing based on the patient referral criteria outlined by the European Malignant Hyperthermia Group (EMHG) [29]. In this study, suspected MH was defined based on the patient referral criteria of the EMHG, including: a family history of MH or unexplained perioperative death; adverse reactions to anesthesia involving signs of hypermetabolism (e.g., unexplained hypercapnia, tachycardia, hyperthermia, muscle rigidity, rhabdomyolysis, or DIC); postoperative or exertional rhabdomyolysis of unknown cause; persistent unexplained hyperCKaemia; exertional heat stroke without predisposing factors; or the presence of a potentially pathogenic RYR1 variant associated with myopathy. In addition, we also included cases that developed postoperative signs and symptoms suggestive of MH, such as hyperthermia, muscle rigidity, elevated creatine kinase levels, and rhabdomyolysis. Informed consent for genetic testing and calcium-induced calcium release (CICR) testing was obtained from all participants or their legal guardians. In cases where using samples already obtained made direct contact with the individual unfeasible, an opt-out approach was used to inform them by posting the research plan on the institution’s website and providing an opportunity for inquiries. The study was approved by the institutional ethics committee (Approval No. E2015-9151-26). A flow chart of patient selection is shown in Figure 1.

Flow diagram showing the selection of 338 individuals from 247 families who underwent gene panel testing based on EMHG referral criteria. Subgroup analyses included CICR-positive and -negative cases and those with CGS grade 5 or 6. CICR (Calcium-Induced Calcium Release): A functional test that measures calcium release from muscle fibers in response to caffeine, used to evaluate MH susceptibility. CGS (Clinical Grading Scale): A scoring system that estimates the likelihood of MH based on clinical signs and labs, ranging from Rank 1 (almost unlikely) to Rank 6 (almost certain).

### 2.2. DNA Extraction

Approximately 5–10 mL of peripheral blood was collected by venipuncture into EDTA-containing tubes. The samples were stored at 4 °C until DNA extraction. Extracted genomic DNA was subsequently stored at −20 °C. When residual muscle tissue obtained for CICR testing was available, DNA was extracted from the muscle tissue. Genomic DNA was extracted from peripheral blood or muscle samples using the QIAamp DNA Blood Mini Kit (Qiagen GmbH, Cologne, Germany) following the manufacturer’s protocol. The concentration and purity of DNA were assessed using a NanoDrop spectrophotometer (Thermo Fisher Scientific, Waltham, MA, USA).

### 2.3. Genetic Panel Testing

Library preparation was performed using the Lotus DNA Library Prep Kit (Integrated DNA Technologies, Coralville, IA, USA) and xGen™ DNA EZ Library Prep Kit (Integrated DNA Technologies, Coralville, IA, USA), following the xGen hybridization capture of DNA libraries (Integrated DNA Technologies, Coralville, IA, USA) protocol. The panel covered 24 genes known to be associated with malignant hyperthermia and calcium regulation in skeletal muscles. The list of 24 investigated genes is shown in Figure 2. Sequencing was conducted on an Illumina platform (Illumina NovaSeq6000; Illumina, San Diego, CA, USA), generating paired-end reads with a target mean coverage of approximately 505× for Lotus DNA Library Prep Kit (Integrated DNA Technologies, Coralville, IA, USA) and 674× for xGen™ DNA EZ Library Prep Kit (Integrated DNA Technologies, Coralville, IA, USA). Quality control of sequencing reads was conducted using FastQC v0.11.5 (Babraham Bioinformatics, Cambridge, UK) to assess sequence quality. Adapter trimming and read filtering were performed using Cutadapt (Freiburg, Germany). Coverage analysis was conducted using mosdepth v0.3.3 (Pedersen and Quinlan, University of Utah, Salt Lake City, UT, USA) to confirm that all target regions achieved a minimum coverage of 100×.

All raw sequencing data obtained from the gene panel testing were submitted to the DNA Data Bank of Japan (DDBJ) Sequence Read Archive (DRA) under accession number BioProject: PRJDB35521.

The 24 genes included in the targeted gene panel, selected for their known or suspected involvement in calcium signaling or malignant hyperthermia susceptibility.

### 2.4. Variant Calling and Filtration

Variant filtration was performed according to GATK (Genome Analysis Toolkit) best practices. BWA was used to align reads to the hg19 reference genome, and variant calling was conducted using GATK HaplotypeCaller (Broad Institute, Cambridge, MA, USA). SNV and indel filtering were performed using the following criteria: SNV filters were QD (Quality by Depth) < 2.0, QUAL (Quality Score) < 30.0, FS (Fisher Strand Bias) > 60.0, MQ (Mapping Quality) < 40.0, MQRankSum < −12.5, ReadPosRankSum (Mapping Quality Rank Sum Test) < −8.0, and SOR (Strand Odds Ratio) > 3.0. Indel filters were QD < 2.0, QUAL < 30.0, FS > 200.0, ReadPosRankSum < −20.0

Annotation was performed using ANNOVAR (version 2018Apr16) and SnpEff/SnpSift v5.1d, incorporating ClinVar for clinical significance classification and the Tohoku Medical Megabank Organization’s Japanese population database (version: 60KJPN) for allele frequency analysis. Additional filtering criteria were applied based on allele frequency in population databases (gnomAD v4.0, ExAC). Variants were classified based on the 2015 ACMG-AMP guidelines [30] into pathogenic, likely pathogenic, variant of uncertain significance (VUS), likely benign, and benign. Pathogenicity was further evaluated using in silico prediction tools, including MutationTaster [31], CADD (GRCh-v1.7) [32], REVEL [33], SIFT [34], and PolyPhen-2 [35].

We applied the ACMG/AMP guideline’s PP3 criterion and included the identified variants in the analysis as supporting pathogenicity if multiple in silico prediction programs classified them as pathogenic. We set the cutoff values as follows: MutationTaster (disease-causing), CADD (score > 20), REVEL (score > 0.5), SIFT (score < 0.05), and PolyPhen-2 (score > 0.5). Specifically, we included the variants in the analysis as candidate pathogenic variants if they were predicted to be pathogenic by at least two out of five in silico prediction programs and had a Minor Allele Frequency (MAF) of less than 0.1% in gnomAD or ExAC. All frameshift variants were considered to be supporting pathogenicity due to their predicted disruptive effects on protein structure and function.

### 2.5. CICR Rate Test

Skeletal muscle specimens were collected via biopsy from the quadriceps or biceps brachii muscles. To create skinned fibers, saponin was used to chemically remove the skeletal muscle cell membranes, and the isometric tension of each sample was measured using a force transducer. The CICR test was conducted following the protocol established by Endo et al. [36,37]. The Ca^2+^ release rates were assessed using solutions with five different Ca^2+^ concentrations (0, 0.3, 1.0, 3.0, and 10.0 μM). The acceleration of CICR rates was evaluated based on our previous research [38]. The mean CICR values were determined from 12 individuals who had tested negative for IVCT and CHCT. A CICR rate exceeding the mean of normal individuals by more than two standard deviations (SD) was defined as an accelerated CICR rate, indicating a predisposition to MH. Patients with an accelerated CICR rate were classified as “CICR-Positive,” meaning their CICR rate was higher than that of the controls, whereas those without acceleration were categorized as “CICR-Negative,” indicating a CICR rate comparable to that of the controls.

### 2.6. Clinical Grading Scale

The MH Clinical Grading Scale (CGS) is a diagnostic tool used to assess the likelihood of MH based on clinical symptoms. It assigns scores ranging from 0 to 88 by summing points for various symptoms, including muscle rigidity, muscle breakdown, respiratory acidosis, hyperthermia, and cardiac abnormalities [39]. Based on the total score, cases are classified into six ranks: a score of 0 (MH Rank 1) suggests that MH is “almost unlikely,” scores between 3 and 9 (MH Rank 2) indicate a “low probability,” scores from 10 to 19 (MH Rank 3) suggest a “somewhat low probability,” scores from 20 to 34 (MH Rank 4) indicate a “somewhat high probability,” scores from 35 to 49 (MH Rank 5) correspond to a “very high probability,” and scores from 50 to 88 (MH Rank 6) indicate that MH is “almost certain.” In this study, we focused on cases with a CGS score of 35 or higher, corresponding to MH Ranks 5 and 6, as these patients are considered to have a strong clinical suspicion of MH. We selected individuals with MH Ranks 5 and 6 to analyze those most likely to be affected by MH.

### 2.7. Data Analysis

Frequencies were calculated based on the number of independent families, i.e., multiple mutation carriers within the same family were counted as one family. Before conducting the subgroup analysis, we first examined the overall distribution of gene mutations detected by the panel test among all patients.

In subgroup analysis 1, we investigated the incidence of gene mutations in the panel test among all CICR-tested patients. Then, in subgroup analysis 2, we focused on a subgroup of patients with high CGS levels and analyzed the incidence of gene mutations within this group.

Statistical analysis was performed using the chi-squared (χ^2^) test in Microsoft Excel. A *p*-value of less than 0.05 was considered statistically significant.

## 3. Results

### 3.1. Patient Characteristics

A total of 338 individuals from 247 independent families were included in this study. Among them, 194 (57.4%) were male and 144 (42.6%) were female. The median age at the time of genetic testing was 38 years (range: 0–91 years). The breakdown of reasons why the subjects underwent genetic panel testing is shown in Figure 3. The most common reason for testing was personal or family history of MH.

Pie chart summarizing the indications for genetic testing: personal history of MH (39%), family history of MH (41%), elevated serum CK levels (5%), and others (15%). MH: malignant hyperthermia, CK: Creatine Kinase.

### 3.2. Genetic Testing Results

Among the 247 families tested, any candidate pathogenic variant were identified in 164 families (66.4%). The RYR1 variant was observed in 94 families (38.1%), the CACNA1S variant in 23 families (9.3%), and variants in other genes in 78 families (31.5%) (Table 1). The distribution of these variants is shown in Figure 4. The distribution of detected variants across the 24 genes in the panel is summarized in Table 2. In our study, we identified 13 mutations in RYR1 and one mutation in CACNA1S that are listed as diagnostic MH mutations on the EMHG website. Detailed information on the variants identified in RYR1 and CACNA1S is provided in Appendix A.

Summary of detected variants among CICR-negative individuals (*n* = 108), showing the distribution and frequency of mutations across the major genes studied. The *p*-values represent the results of chi-square tests comparing the detection frequencies of RYR1, CACNA1S, and other variants between patients with CGS ranks 5 and 6 and those who were CICR-positive. No significant differences were observed between the two groups.

Number of families with pathogenic or likely pathogenic variants identified in RYR1, CACNA1S, or other genes. RYR1: Ryanodine Receptor 1, CACNA1S: Calcium Voltage-Gated Channel Subunit Alpha1 S.

### 3.3. Correlation Between Genetic Findings and CICR Test Results

The CICR test was performed on 239 individuals (210 families). Among them, 131 (54.8%) showed an accelerated CICR rate and were classified as “CICR-Positive”, while 108 (45.2%) were classified as “CICR-Negative.” The proportion of CICR-Positive families harboring pathogenicity-supporting RYR1 variants was 55 families (42.0%), while 7 families (5.3%) of CICR-Negative families carried such variants. The proportion of CICR-Positive families harboring pathogenicity-supporting CACNA1S variants was 7 families (5.3%), while 7 families (6.5%) of CICR-Negative families carried such variants (Table 1). Appendix A summarizes the CICR test results and the associated variants.

### 3.4. Clinical Grading Scale (CGS) and Genetic Findings

There were 91 patients clinically suspected of malignant hyperthermia (MH) due to general anesthesia. Among them, 50 patients had a CGS score of 35 or higher (CGS Ranks 5 and 6), 22 had a score below 35 (CGS Ranks 1 to 4), and the remaining 19 were unassessable. The proportion of CGS Ranks 5 and 6 families harboring pathogenicity-supporting RYR1 variants was 28 families (56.0%), while 5 families (27.3%) of CGS Ranks 1 to 4 families carried such variants. The proportion of CGS Ranks 5 and 6 families harboring pathogenicity-supporting CACNA1S variants was 6 families (12.0%), while 4 families (18.2%) of CGS Ranks 1 to 4 families carried such variants (Table 1).

RYR1 variants were identified in 7 of 18 patients with a CGS score of 5, and in 21 of 32 patients with a CGS score of 6 (*p* = 0.13). The distribution of RYR1 variants among patients with CGS scores of 5 and 6 is shown in Appendix A.

## 4. Discussion

In this study, we conducted genetic panel testing on Japanese patients with MH or suspected MH, as well as their family members, to assess the frequency of pathogenic variants in *RYR1* and *CACNA1S*, and to identify potentially novel variants in other genes related to calcium regulation. Variants with high pathogenic potential were identified in 29.8% of families for *RYR1*, 6.5% for *CACNA1S*, and 25.3% in other genes. These results support the well-established role of *RYR1* as the major MH susceptibility gene. At the same time, the high proportion (25.4%) of variants found in genes other than *RYR1* and *CACNA1S* suggests that MH susceptibility may involve a broader and more diverse genetic background. We also conducted subgroup analyses focusing on two clinically relevant cohorts: individuals with a positive CICR test and those with high scores on the Clinical Grading Scale (CGS). In the CICR-positive group, *RYR1* and *CACNA1S* pathogenic variants were found in 42.0% and 5.3% of individuals, respectively. In contrast, in the group with CGS ranks 5 or 6, *RYR1* and *CACNA1S* variants were identified in 56.0% and 12.0% of individuals, respectively. These findings suggest that *RYR1* variants are more likely to be present in patients with strong clinical or laboratory-based evidence of MH. While the reported prevalence of *CACNA1S* variants in MH cohorts is generally around 1–2%, our study identified *CACNA1S* variants in nearly 10% of Japanese MH patients [40,41]. This finding suggests that *CACNA1S* variants may be more frequent in the Japanese population compared to previous reports from other populations.

Previous studies have reported detection rates of pathogenic *RYR1* variants in MH-susceptible individuals ranging from 60% to 86%, depending on the population and methodology used [15,16,17,18,19]. In our study, the overall detection rate of variants with pathogenic potential was 48.2%, and *RYR1* variants were observed in 29.8% of families, which is somewhat lower than the higher end of the range reported in Western cohorts. Even when focusing on individuals with high CGS scores or CICR-positive results, the detection rates of pathogenic RYR1 variants were 56% and 42%, respectively—both lower than those reported in previous studies. The lower detection rate of RYR1 variants in the Japanese cohort compared to Western cohorts may be attributed to several factors. One possible explanation is ethnic differences in the genetic architecture of MH susceptibility, with non-RYR1 genes potentially playing a larger role in East Asian populations. For example, CACNA1S variants were observed at a relatively higher frequency in our Japanese cohort, suggesting such population-specific variation. In addition, the CICR assay commonly used in Japan may identify MH susceptibility caused by mechanisms other than RYR1 mutations, which could contribute to the lower detection rate of RYR1-related MH susceptibility among CICR-positive individuals. While North America and Europe have established organizations such as the Malignant Hyperthermia Association of the United States (MHAUS) [42] and the European Malignant Hyperthermia Group (EMHG) [19], dedicated to MH research, Asia is still developing regional collaboration efforts, such as the Asian Malignant Hyperthermia Alliance (AMHA) [43]. Although broad-scale genetic screening of the general population in Singapore has identified MH-related gene variants [44], large-scale genetic studies specifically targeting MH patients have not yet been conducted in most parts of Asia. To clarify potential ethnic differences in MH susceptibility, further genetic analyses focused on clinically diagnosed MH patients across different Asian populations are warranted.

In addition, several potentially pathogenic variants were identified in genes other than *RYR1* [8,45] and *CACNA1S* [40,46,47], STAC3 [48] consistent with previous reports suggesting a polygenic or multifactorial basis for MH. However, the clinical relevance of these variants remains unclear. Functional validation through cellular assays and animal models will be essential to determine whether these novel variants contribute to MH susceptibility. Among the candidate genes, *CACNB1*, *CASQ1*, *SERCA1*, *CASQ2*, and *KCNA1* have been reported to be potentially related to the mechanism of MH [49,50], and several pathogenic variants were found in our analysis. *CACNB1* encodes the β1 subunit of the dihydropyridine receptor, which modulates calcium influx and excitation–contraction coupling [51]. *CASQ1* and *CASQ2* are calsequestrins that serve as calcium-binding proteins in the sarcoplasmic reticulum of skeletal and cardiac muscle, respectively, playing key roles in calcium storage and release [52]. *ATP2A1* encodes the SERCA1 pump, essential for sarcoplasmic reticulum calcium reuptake during muscle relaxation [53]. *KCNA1* encodes a voltage-gated potassium channel that regulates membrane excitability in skeletal muscle [50]. Given their involvement in calcium homeostasis, excitation–contraction coupling, and muscle excitability, these proteins may possibly be associated with MH susceptibility. In addition, *ASPH* [54] and *TRPV1* [55] have been reported to be associated with calcium dysregulation or skeletal muscle excitability, and further investigation is also required. Specifically, ASPH may affect excitation–contraction coupling through post-translational modification of key proteins, while TRPV1 has been shown to modulate intracellular calcium levels in skeletal muscle.

Although the variants identified in this study were predicted to be potentially pathogenic based on in silico analyses, their direct involvement in the pathophysiology of MH remains unclear. Future studies employing functional assays to assess abnormalities in calcium release and muscle contractility will be essential to clarify the pathogenic roles of these variants [56,57]. Furthermore, to better understand the genetic diversity of MH susceptibility in East Asian populations, it is crucial to conduct collaborative multicenter and multinational studies for comprehensive case collection and analysis. If future functional validation confirms the involvement of the newly identified variants in MH, our findings could contribute to improved screening strategies and the expansion of MH susceptibility panels, particularly tailored for East Asian populations.

This study has several limitations. First, the cohort was limited to cases collected from a single institution, and unreported or undetected cases may have been missed. Second, while in silico prediction tools were used to evaluate the pathogenicity of variants, no functional validation experiments were conducted. This is particularly relevant for variants of uncertain significance (VUS), whose clinical implications may become evident with future functional studies. Third, the gene panel used in this study did not encompass all MH-related genes, and it is possible that unknown causative genes may have contributed to MH susceptibility in some cases.

## 5. Conclusions

In this study, we performed gene panel testing on Japanese patients with confirmed or suspected malignant hyperthermia (MH) and characterized the distribution of variants supporting pathogenicity in *RYR1* and *CACNA1S*, along with the identification of several novel variants. Notably, the frequency of *CACNA1S* variants supporting pathogenicity appeared to be higher in our Japanese cohort compared to Western populations. These findings improve our understanding of the genetic background of MH in Japanese and other Asian populations and provide a foundation for future functional studies and clinical applications.

## Figures and Tables

**Figure 1 genes-16-00944-f001:**
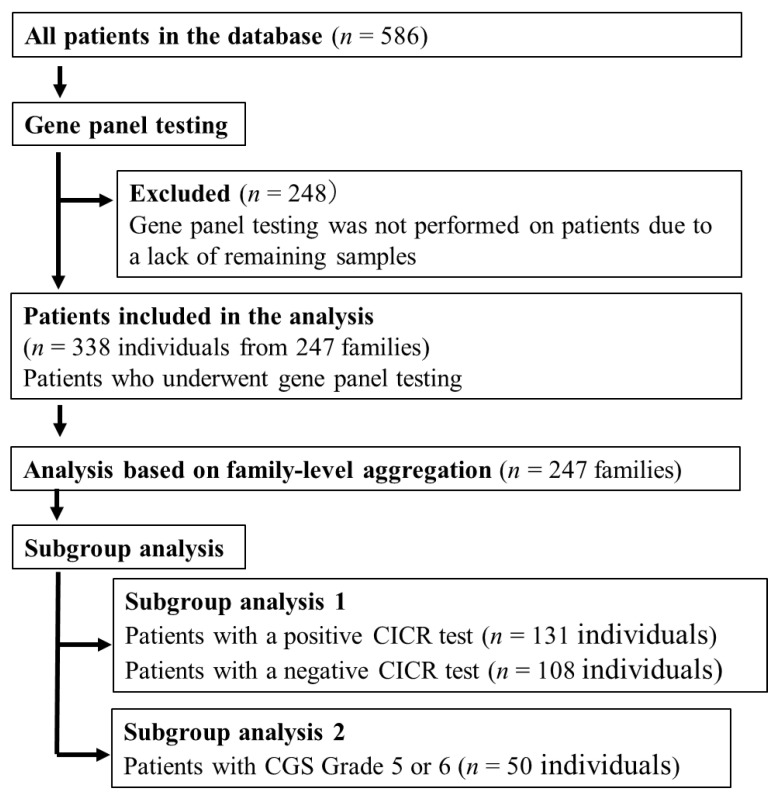
Patient selection criteria.

**Figure 2 genes-16-00944-f002:**
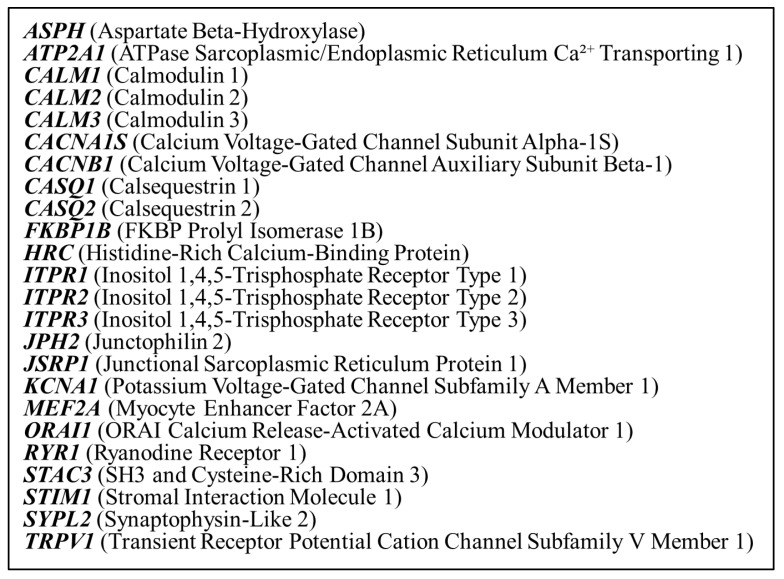
Investigated genes in the gene panel.

**Figure 3 genes-16-00944-f003:**
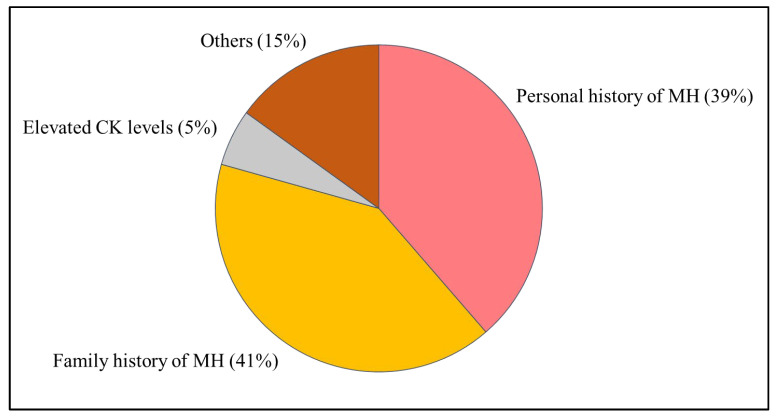
Reasons for undergoing genetic testing.

**Figure 4 genes-16-00944-f004:**
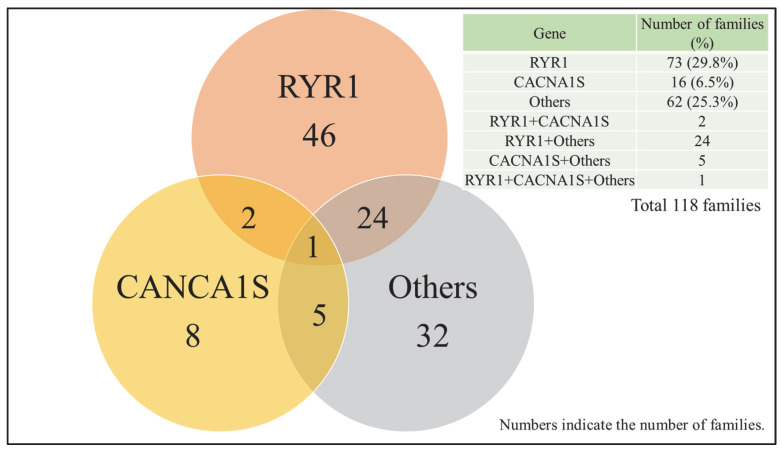
Distribution of gene variants with potential pathogenicity.

**Table 1 genes-16-00944-t001:** Frequency of genetic variants in RYR1, CACNA1S, and other genes.

	No of Families	RYR1, *n* (%)	CACNA1S, *n* (%)	Others, *n* (%)			
Over All	245	73 (29.8%)	16 (6.5%)	62 (25.3%)			
	No of families	RYR1	CACNA1S	Others	*p*-value (RYR1)	*p*-value (CACNA1S)	*p*-value (Others)
CGS Ranks 5 and 6	50	28 (56.0%)	6 (12.0%)	15 (30.0%)	0.091	0.137	0.737
CICR-Positive	131	55 (42.0%)	7 (5.3%)	34 (27.5%)			
CICR-Negative	108	11 (10.2%)	7 (6.5%)	18 (15.7%)			

**Table 2 genes-16-00944-t002:** Detected genetic variants across the 24 investigated genes.

RYR1			CACNA1S	Others	
R14W	P1592L	R3321S	N64D	**E200K (ASPH)**	R228Q (ITPR3)
D17N	W1625C	**K3367R**	**R174W ^†^**	K350R (ASPH)	R527H (ITPR3)
T84M	**R2163H**	N3555I	L298I	R369C (ASPH)	T808I (ITPR3)
**Q155K**	**V2168M**	V3602M	G321V	D27G (ATP2A1)	F1750L (ITPR3)
**R163C**	**T2206M**	N3913D	A560T	I298S (ATP2A1)	V1867M (ITPR3)
D167G	M2208K	Q3964Rfs*105	S879P	R134P (ATP2A1)	S2045G (ITPR3)
E176D	**V2280I**	L3984P	F1060V	R534W (ATP2A1)	V2386I (ITPR3)
E209K	**R2336H**	I4184M	F1161L ^†^	P571L (ATP2A1)	R436C (JPH2)
R274H	**S2345R**	K4477N	D1382V	P571R (ATP2A1)	K249_E253del (JSRP1)
A291V	S2345T	L4769F	G1459S	R107H (CALM2)	Q420_P421insQQQ (MEF2A)
R316L	I2358T	**L4838V**		S279L (CACNB1)	R287C (ORAI1)
**G341R**	P2366R	Y4852S		I164M (CASQ1)	S61Lfs* (ORAI1)
K364R	D2431E	I4898T		A196V (CASQ1)	L30fs (STAC3)
R367Q	D2431H	W5020G		G347* (CASQ1)	H89Lfs*20 (STAC3)
R391C	R2454G	A5025G		D261_V262insDD (HRC)	G142S (STAC3)
**Y522C**	**R2508C**			E376* (HRC)	T502dup (STIM1)
**R530H**	**R2508H**			H9Lfs*17 (HRC)	R522S (STIM1)
R533H	R2615H			T310A (ITPR2)	G586V (STIM1)
S604P	R2625C			D723N (ITPR2)	A115T (SYPL2)
**R614L**	D2769N			I1762T (ITPR2)	E211K (TRPV1)
R727L	E2824K			E975A (ITPR2)	D297N (TRPV1)
R870W	**E3104K**			T969I (ITPR2)	Q261P (TRPV1)
E887K	R3119H			A41V (ITPR3)	R772C (TRPV1)
N1164H	M3266I			A147T (ITPR3)	V508M (TRPV1)

Comprehensive list of variants identified through gene panel testing, including those in RYR1, CACNA1S, and 22 other candidate genes. EMHG-recognized diagnostic variants are shown in red. Variants shown in bold are classified as pathogenic or likely pathogenic based on ClinVar annotation. ^†^ Homozygous variant.

## Data Availability

The datasets used and/or analyzed during the current study are available from the corresponding author on reasonable request.

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
