# Peer review of "Genetic Panel Testing for Malignant Hyperthermia in Japan: Discovery of Novel Variants and Clinical Implications"

_genes, 2025, doi:10.3390/genes16080944_

Round 1

Reviewer 1 Report

Comments and Suggestions for Authors

Overall, nice presentation. Here are my comments.Review of "Genetic panel testing for malignant hyperthermia in Japan: discovery of novel variants and clinical implications"

1. Clinical correlation analysis requires strengthening:
-Please provide a supplementary table detailing all Clinical Grading Scale (CGS) scores with their corresponding rankings
-What specific clinical criteria defined "suspected" MH beyond the EMHG referral guidelines?
-Missing data on triggering agents: The absence of data on specific anesthetic agents that triggered MH episodes is a significant omission. Different volatile anesthetics (sevoflurane, desflurane, isoflurane) and succinylcholine may have varying triggering potentials, and correlating this information with genetic variants would be valuable.

2. Modest genetic detection rates warrant further discussion: The relatively low detection rates (42% RYR1 in CICR-positive, 56% in CGS ranks 5-6) suggest either incomplete genetic understanding, which would lead to limited clinical applicability.
-For the Supplementary Table:
--Add a "Header Explanation" sheet clarifying column headers and the meaning of blue highlighting
--Provide: number of families (column J), number of individuals positive for each RYR1/CACNA1S variant, and crucially, the number of alleles detected to determine heterozygosity status (column L is unclear regarding heterozygosity)
--Include ClinVar links where available (into additional column).
3. Analysis of multiple gene variants needed: Your data suggests ~31 families (19% of variant-positive families) harbor variants in ≥2 genes, indicating possible oligogenic inheritance. Please add a column to the Supplementary Table identifying co-occurring variants in other genes.

4. Maybe I missed it - did you previously publish the information about how these genes on the panel were chosen? 

Author Response

________________________________________

Reviewer 1

Overall, nice presentation. Here are my comments. Review of "Genetic panel testing for malignant hyperthermia in Japan: discovery of novel variants and clinical implications"

Response:

We sincerely thank Reviewer 1 for the positive overall evaluation of our manuscript. We greatly appreciate your constructive and insightful comments, which have been extremely helpful in clarifying our clinical and genetic analyses. Your feedback allowed us to improve the scientific rigor and presentation of our work, and we have addressed all points carefully in the revised version.

During the course of our revisions, we noticed that one family had been inadvertently omitted from the count. As a result, we have corrected the corresponding numbers in both the abstract and the main text. The revised sections are highlighted in red and documented in the revision history. We sincerely apologize for this oversight.

  1. Clinical correlation analysis requires strengthening:

-Please provide a supplementary table detailing all Clinical Grading Scale (CGS) scores with their corresponding rankings

Response:

Thank you for this helpful suggestion. We have added a new Supplementary Table (Supplementary Table 2) listing all individuals with their respective Clinical Grading Scale (CGS) scores and corresponding MH ranks. This table allows readers to clearly understand the clinical classification of each individual.

There was no significant difference in the frequency of RYR1 variants between patients with CGS grades 5 and 6. We have added a description of the statistical method in the Methods section (2.7 Data Analysis) and included this result in the Results section (3.4 Clinical Grading Scale (CGS) and Genetic Findings).

-What specific clinical criteria defined "suspected" MH beyond the EMHG referral guidelines?

Response:

Thank you for your comment. In addition to the EMHG referral guidelines, we also included cases that developed postoperative signs and symptoms suggestive of MH, such as hyperthermia, muscle rigidity, elevated creatine kinase levels, and rhabdomyolysis. We have added this information to the manuscript (Page 2, Lines 81–91).

-Missing data on triggering agents: The absence of data on specific anesthetic agents that triggered MH episodes is a significant omission. Different volatile anesthetics (sevoflurane, desflurane, isoflurane) and succinylcholine may have varying triggering potentials, and correlating this information with genetic variants would be valuable.

Response:

We have now added a column to Supplementary Table 2 showing the known anesthetic agents used during the suspected MH episodes. We appreciate the reviewer’s comment regarding the missing data on the specific anesthetic agents that triggered MH episodes. We agree that correlating anesthetic exposure with genetic variants could provide additional insights. However, in our cohort, for the subset of patients in whom exposure data were available, there did not appear to be any clear bias in the distribution of anesthetic agents (e.g., sevoflurane, desflurane, isoflurane, or succinylcholine) with respect to specific genetic variants.

  1. Modest genetic detection rates warrant further discussion: The relatively low detection rates (42% RYR1 in CICR-positive, 56% in CGS ranks 5-6) suggest either incomplete genetic understanding, which would lead to limited clinical applicability.

-For the Supplementary Table:

--Add a "Header Explanation" sheet clarifying column headers and the meaning of blue highlighting

--Provide: number of families (column J), number of individuals positive for each RYR1/CACNA1S variant, and crucially, the number of alleles detected to determine heterozygosity status (column L is unclear regarding heterozygosity)

--Include ClinVar links where available (into additional column).

Response:

We thank the reviewer for this suggestion.

-We have added a new sheet titled “Header Explanation” to the Supplementary Table to clarify the meaning of each column header as well as the significance of blue highlighting.

-The numbers of families, the number of individuals carrying each RYR1 or CACNA1S variant, and the number of alleles detected have been explicitly indicated. We have also clarified the zygosity status to distinguish between heterozygous and homozygous cases.

-Additionally, a new column has been added to provide ClinVar links, where available, for each variant.

  1. Analysis of multiple gene variants needed: Your data suggests ~31 families (19% of variant-positive families) harbor variants in ≥2 genes, indicating possible oligogenic inheritance. Please add a column to the Supplementary Table identifying co-occurring variants in other genes.

Response:

Thank you for highlighting this important point. We agree that potential oligogenic inheritance is an important aspect. Since Supplementary Table 1 focuses on variant-level information, it does not allow us to represent situations where multiple variants are present within a single individual. However, in our revision of the manuscript, we have added Supplementary Tables 2 and 3, which include the CGS and CICR results, respectively. These new tables provide information on individuals carrying multiple variants.

  1. Maybe I missed it - did you previously publish the information about how these genes on the panel were chosen?

Response:
We apologize for the lack of clarity. The genes included in our targeted panel are described in the third paragraph of the Discussion. As this was an exploratory study, the panel includes some genes that have not been previously reported to be associated with MH. Some explanations have been added to the third paragraph of the Discussion regarding this revision.

Reviewer 2 Report

Comments and Suggestions for Authors
  1. Introduction:

Please present epidemiological data on the incidence of malignant hyperthermia (MH), including rates in North America, Europe, and Asia, along with relevant references.

Briefly describe the functions of the proteins encoded by the RYR1 and CACNA1S genes and their role in the pathophysiology of MH. Explain how pathogenic variants in these genes disrupt calcium homeostasis and lead to the clinical manifestations of MH.

Please indicate the chromosomal location of RYR1 and CACNA1S.

  1. Materials and Methods

The sentence "Patients and/or their families with a history of suspected malignant hyperthermia or with suspected malignant hyperthermia were included in our study" should be expanded upon. Please explain the basis for determining MH susceptibility. Briefly outline the diagnostic criteria for MH susceptibility according to the European Malignant Hyperthermia Group (EMHG) guidelines.

Were any patients who developed MH among the study participants?

Although the abbreviation CICR is explained in the abstract, the full term name should be provided in the main text upon first use.

Please define the abbreviation CGS (Clinical Grading Scale) in the manuscript text and briefly describe this scale and its clinical significance for assessing the likelihood of susceptibility to MH. Please also carefully review and define all other abbreviations throughout the text (including figures, tables, and descriptions/legends).

lease provide information on the volume of blood collected, the method of collection, how it was stored before DNA extraction, and how DNA samples were preserved before analysis.

  1. Results

Figure 2: Please ensure that all gene symbols are presented according to HGNC nomenclature standards.

Figure 3: The figure layout should be reformatted to prevent text from overlapping with graphic elements.

Table 1:

Please provide the results of statistical analyses comparing the frequencies of genetic variants in the RYR1 and CACNA1S genes in different subgroups. Please add a header row specifying the type of analysis or measured variable, regardless of the table title.

Table 2:

Please explain why most variants were described at the protein level and some at the cDNA level. Please use a consistent notation system.

Please indicate whether the variants were detected in homozygous or heterozygous form.

Please highlight variants classified as pathogenic or likely pathogenic (e.g., by bolding them or using symbols).

Figure 4:

The legend refers to text in red, but this highlighting is not visible in the figure. Please correct the text accordingly.

In addition, please provide the percentage of families in which each variant was identified.

  1. Discussion

The first two sentences of the discussion appear to be editorial guidelines and should be removed. The statement, "Previous studies have shown that detection rates of pathogenic variants in individuals susceptible to MH ranged from 50% to 70%, depending on the population and methodology used," should be supported by appropriate literature references.

Please discuss the potential reasons for the lower detection rate of RYR1 variants in the Japanese cohort compared to Western cohorts. Please briefly discuss this topic.

The authors note: "Furthermore, our study identified several potentially pathogenic variants in genes other than RYR1 and CACNA1S, including STAC3, which is consistent with previous reports suggesting a polygenic or multifactorial basis for MH – please provide relevant citations to these previous reports.

Regarding candidate genes such as CACNB1, CASQ1, ATP2A1 (SERCA1), CASQ2, and KCNA1, please provide a few sentences explaining the rationale for considering these genes as candidate genes – i.e., potentially functionally relevant to MH (e.g., role in calcium metabolism, excitation-contraction coupling, or muscle excitability).

Similarly, please discuss the possible involvement of ASPH and TRPV1 in the pathomechanisms associated with MH (e.g., evidence linking them to abnormal calcium signaling or skeletal muscle excitability).

Please specify the potential clinical implications of your results." research, such as improved screening strategies, expanded MH susceptibility panels or identification of new targets for functional validation.

Author Response

Our responses to the reviewers’ comments are as follow:

Reviewers' comments and our responses are listed below. The changes are presented in red character in the revised text.

________________________________________

Reviewer 2

Response:
We thank Reviewer 2 for the detailed and constructive feedback, which has significantly improved the clarity and quality of our manuscript. Below, we provide point-by-point responses to each comment.

During the course of our revisions, we noticed that one family had been inadvertently omitted from the count. As a result, we have corrected the corresponding numbers in both the abstract and the main text. The revised sections are highlighted in red and documented in the revision history. We sincerely apologize for this oversight.

Introduction:

Please present epidemiological data on the incidence of malignant hyperthermia (MH), including rates in North America, Europe, and Asia, along with relevant references.

Response:
Thank you for your valuable comment. We have added epidemiological data in the Introduction (page 1, lines 32–35). In accordance with this revision, we have also added three references.

Briefly describe the functions of the proteins encoded by the RYR1 and CACNA1S genes and their role in the pathophysiology of MH. Explain how pathogenic variants in these genes disrupt calcium homeostasis and lead to the clinical manifestations of MH.

Response:
We appreciate your helpful suggestion. In response, we have included descriptions of the RYR1 and CACNA1S genes in the Introduction (page 1, lines 35–44).

Please indicate the chromosomal location of RYR1 and CACNA1S.

Response:
Thank you for your comment. We have added the chromosomal locations of RYR1 and CACNA1S in the Introduction (page 1, lines 36 and 40, respectively).

Materials and Methods

The sentence "Patients and/or their families with a history of suspected malignant hyperthermia or with suspected malignant hyperthermia were included in our study" should be expanded upon. Please explain the basis for determining MH susceptibility. Briefly outline the diagnostic criteria for MH susceptibility according to the European Malignant Hyperthermia Group (EMHG) guidelines.

Response:
Thank you for your comment. We have added an explanation of "suspected MH" in the Materials and Methods section (page 2, lines 79–88), including the basis for determining MH susceptibility and the diagnostic criteria according to the European Malignant Hyperthermia Group (EMHG) guidelines.

Were any patients who developed MH among the study participants?

Response:
Yes, some of the study participants had experienced malignant hyperthermia (MH) episodes. In response to Reviewer 1’s comment, we have listed the patients who developed MH and included their CGS scores and grades in Supplementary Table 2.

Although the abbreviation CICR is explained in the abstract, the full term name should be provided in the main text upon first use.

Response:
Thank you for your comment. We have revised the manuscript to include the full term for CICR upon its first use in the main text.

Please define the abbreviation CGS (Clinical Grading Scale) in the manuscript text and briefly describe this scale and its clinical significance for assessing the likelihood of susceptibility to MH. Please also carefully review and define all other abbreviations throughout the text (including figures, tables, and descriptions/legends).

Response:
Thank you for your comment. We have already described the Clinical Grading Scale (CGS) in the Materials and Methods section (2.6 Clinical Grading Scale). However, to improve clarity, we have now defined the abbreviation “CGS” upon its first appearance in the main text. We have also carefully reviewed the manuscript, including figures, tables, and legends, to ensure that all abbreviations are properly defined.

Please provide information on the volume of blood collected, the method of collection, how it was stored before DNA extraction, and how DNA samples were preserved before analysis.

Approximately 5 mL of blood was collected by venipuncture for DNA extraction and stored in collection tubes under refrigerated or frozen conditions.

Response:
Thank you for your comment. We have added detailed information regarding the volume of blood collected, collection method, storage conditions before DNA extraction, and preservation of DNA samples in the Materials and Methods section (2.2 DNA Extraction).

Results

Figure 2: Please ensure that all gene symbols are presented according to HGNC nomenclature standards.

Response:
Thank you for your comment. We have corrected the gene symbol SERCA1 to ATP2A1 and formatted it in italics in Figure 2 to comply with HGNC nomenclature standards.

Figure 3: The figure layout should be reformatted to prevent text from overlapping with graphic elements.

Response:
Thank you for your comment. We have adjusted the layout of Figure 3 to prevent text from overlapping with graphic elements.

Table 1:

Please provide the results of statistical analyses comparing the frequencies of genetic variants in the RYR1 and CACNA1S genes in different subgroups. Please add a header row specifying the type of analysis or measured variable, regardless of the table title.

Response:
Thank you for your valuable comment. We have organized the table and added a header row specifying the variables measured. Additionally, we performed statistical analyses comparing the frequencies of genetic variants in the RYR1 and CACNA1S genes, as well as other variants, across different subgroups using the chi-square test. The results, including p-values, have been added to Table 1 for clarity.

Table 2:

Please explain why most variants were described at the protein level and some at the cDNA level. Please use a consistent notation system.

Response:
Thank you for your comment. For the p.Gly341Arg amino acid variant, we had originally included cDNA information because multiple codon changes can lead to this variant. However, since cDNA information is now provided in the supplementary table, we have chosen to display only the amino acid change in Table 2.

Please indicate whether the variants were detected in homozygous or heterozygous form.

Response:
Thank you for your comment. We have added information on homozygous and heterozygous variants to Table 2. To keep the table clear, we marked only the homozygous variants with a † symbol and included an explanation in the footnote.

Please highlight variants classified as pathogenic or likely pathogenic (e.g., by bolding them or using symbols).

Response:
Thank you for your comment. Variants classified as pathogenic or likely pathogenic are shown in bold.

Figure 4:

The legend refers to text in red, but this highlighting is not visible in the figure. Please correct the text accordingly.

Response:
Thank you for your comment. This was our mistake. We have removed the statement regarding the use of red text.

In addition, please provide the percentage of families in which each variant was identified.

Response:
Thank you for your comment. We have added percentages to the table on the right side of the figure.

Discussion

The first two sentences of the discussion appear to be editorial guidelines and should be removed. The statement, "Previous studies have shown that detection rates of pathogenic variants in individuals susceptible to MH ranged from 50% to 70%, depending on the population and methodology used," should be supported by appropriate literature references.

Response:
Thank you for your comment. The sentence in question was mistakenly left in the manuscript due to our oversight.

We have deleted the following text:

“Authors should discuss the results and how they can be interpreted from the perspective of previous studies and of the working hypotheses. The findings and their implications should be discussed in the broadest context possible. Future research directions may also be highlighted.”

Regarding the detection rate of RYR1 variants, we have added relevant references to the section you pointed out. We have also revised the Introduction accordingly.

Please discuss the potential reasons for the lower detection rate of RYR1 variants in the Japanese cohort compared to Western cohorts. Please briefly discuss this topic.

Response:
Thank you for your comment. We have added the following sentence to the second paragraph of the Discussion section. “Even when focusing on individuals with high CGS scores or CICR-positive results, the detection rates of pathogenic RYR1 variants were 56% and 42%, respectively—both lower than those reported in previous studies. The lower detection rate of RYR1 variants in the Japanese cohort compared to Western cohorts may be attributed to several factors. One possible explanation is ethnic differences in the genetic architecture of MH sus-ceptibility, with non-RYR1 genes potentially playing a larger role in East Asian popu-lations. For example, CACNA1S variants were observed at a relatively higher frequency in our Japanese cohort, suggesting such population-specific variation. In addition, the CICR assay commonly used in Japan may identify MH susceptibility caused by mech-anisms other than RYR1 mutations, which could contribute to the lower detection rate of RYR1-related MH susceptibility among CICR-positive individuals.”

The authors note: "Furthermore, our study identified several potentially pathogenic variants in genes other than RYR1 and CACNA1S, including STAC3, which is consistent with previous reports suggesting a polygenic or multifactorial basis for MH – please provide relevant citations to these previous reports.

Response:
Thank you for your comment. We apologize, but we were unable to identify the sentence you pointed out in the manuscript. The following article provides an explanation regarding the multifactorial basis of malignant hyperthermia:

Miller DM, Daly C, Aboelsaod EM, et al. Genetic epidemiology of malignant hyperthermia in the UK. Br J Anaesth. 2018 Oct;121(4):944-952. doi:10.1016/j.bja.2018.06.028.

We are ready to revise the manuscript once we understand the specific part you are referring to.

Regarding candidate genes such as CACNB1, CASQ1, ATP2A1 (SERCA1), CASQ2, and KCNA1, please provide a few sentences explaining the rationale for considering these genes as candidate genes – i.e., potentially functionally relevant to MH (e.g., role in calcium metabolism, excitation-contraction coupling, or muscle excitability).

Response:
Thank you for your suggestion. We have added the following explanation.

CACNB1 encodes the β1 subunit of the dihydropyridine receptor, which modulates calcium influx and excitation–contraction coupling. CASQ1 and CASQ2 are calsequestrins that serve as calcium-binding proteins in the sarcoplasmic reticulum of skeletal and cardiac muscle, respectively, playing key roles in calcium storage and release. ATP2A1 encodes the SERCA1 pump, essential for sarcoplasmic reticulum calcium reuptake during muscle relaxation. KCNA1 encodes a voltage-gated potassium channel that regulates membrane excitability in skeletal muscle. Given their involvement in calcium homeostasis, excitation–contraction coupling, and muscle excitability, these proteins may possibly be associated with MH susceptibility.

Similarly, please discuss the possible involvement of ASPH and TRPV1 in the pathomechanisms associated with MH (e.g., evidence linking them to abnormal calcium signaling or skeletal muscle excitability).

Response:
Thank you for your comment. We have added the following sentence to the third paragraph of the Discussion section.

Specifically, ASPH may affect excitation–contraction coupling through post-translational modification of key proteins, while TRPV1 has been shown to modulate intracellular calcium levels in skeletal muscle.

Please specify the potential clinical implications of your results." research, such as improved screening strategies, expanded MH susceptibility panels or identification of new targets for functional validation.

Response:
We appreciate your suggestion to elaborate on the potential clinical implications of our findings. In response, we have revised the fourth paragraph of the Discussion section to include the following sentence:

If future functional validation confirms the involvement of the newly identified variants in MH, our findings could contribute to improved screening strategies and the expansion of MH susceptibility panels, particularly tailored for East Asian populations.

________________________________________

Round 2

Reviewer 1 Report

Comments and Suggestions for Authors

I reviewed revised manuscript and accept it.

Reviewer 2 Report

Comments and Suggestions for Authors

The authors have carefully revised their manuscript, which is now ready for publication.